# Design and Experimental Study of a Novel Semi-Physical Unmanned-Aerial-Vehicle Simulation Platform for Optical-Flow-Based Navigation

Zhonglin Lin [1], Weixiong Wang [1], Yufeng Li [1], Xinglong Zhang [2], Tianhong Zhang [2], Haitao Wang [1], Xianyu Wu [1] and Feng Huang [1,*]

1 School of Mechanical Engineering and Automation, Fuzhou University, Fuzhou 350108, China
2 Jiangsu Province Key Laboratory of Aerospace Power System, College of Energy and Power Engineering, Nanjing University of Aeronautics and Astronautics, Nanjing 210016, China
* Correspondence: huangf@fzu.edu.cn; Tel.: +86-151-9597-5633

**Abstract:** In the process of unmanned aerial vehicle (UAV) visual-navigation-algorithm design and accuracy verification, the question of how to develop a high-precision and high-reliability semi-physical simulation platform has become a significant engineering problem. In this study, a new UAV semi-physical-simulation-platform architecture is proposed, which includes a six-degree-of-freedom mechanical structure, a real-time control system and real-time animation-simulation software. The mechanical structure can realistically simulate the flight attitude of a UAV in a three-dimensional space of $4 \times 2 \times 1.4$ m. Based on the designed mechanical structure and its dynamics, the control system and the UAV real-time flight-animation simulation were designed. Compared with the conventional simulation system, this system enables real-time flight-attitude simulation in a real physical environment and simultaneous flight-attitude simulation in virtual-animation space. The test results show that the repeated positioning accuracy of the three-axis rotary table reaches $0.006°$, the repeated positioning accuracy of the three-axis translation table reaches $0.033$ mm, and the dynamic-positioning accuracy reaches $0.04°$ and $0.4$ mm, which meets the simulation requirements of high-precision visual UAV navigation.

**Keywords:** unmanned aerial vehicle; visual navigation; six degrees of freedom; semi-physical simulation platform; optical flow

## 1. Introduction

In recent years, with the increasing need for special photography in film and television, agriculture, firefighting, rescue and other fields, multi-rotor unmanned aerial vehicles (UAV) have been widely used in practice due to their reliability and stability, small size, and ease of handling [1–3]. With the widespread application of multi-rotor UAVs, multi-rotor-UAV navigation technology has also gradually attracted the attention of various research institutions. Due to the small size and light weight of multi-rotor UAVs, it is difficult to use larger navigation devices to multi-rotor UAVs; therefore, the current multi-rotor UAV navigation mostly uses a combination of GPS and inertial navigation system (INS) navigation [4–6]. However, GPS navigation is essentially a type of radio navigation that inevitably communicates with the outside world. Due to the instability of the GPS signal, multi-rotor-UAV crashes occur from time to time. Meanwhile, visual-navigation technology based on optical flow directly processes continuous image sequences by transforming the optical-flow-field information calculated by the optical flow method directly into motion-field information through a mathematical model. The navigation system using the optical-flow method is easy to implement and the real-time performance is guaranteed, which has become an important research direction for visual navigation. With the development of microelectronics, CMOS cameras are moving toward miniaturization,

and the development of optical-flow-navigation technology has therefore reached a new stage. In 2017, McGuire, K. et al. [7] implemented optical-flow autonomous navigation on a 40-gram UAV. In 2020, Back, S. et al. [8] used a neural-network-based optical-flow algorithm to track bicycle-path-obstacle avoidance, which enables the UAV to handle various situations encountered while traveling on a fixed track by combining route tracking, interference recovery, and obstacle avoidance. The feasibility of the method was also verified using dataset simulations and actual flight experiments.

To solve the problems of low accuracy and poor stability of most existing optical-flow-method navigation algorithms, a semi-physical simulation platform for UAV visual navigation needs to be designed and developed before outdoor flight tests of the visual-navigation algorithms can be conducted. The semi-physical-simulation-test validation will be conducted through this platform to improve the algorithm's iteration speed while avoiding flight accidents and economic losses in the real outdoor tests caused by algorithm errors. When the stability, real-time and accuracy of the optical-flow-method navigation algorithm meet the test-flight requirements, the actual flight test will be conducted.

The existing simulation tests are mainly divided into three categories: numerical computer simulation, hardware-in-the-loop (HIL) simulation and semi-physical simulation [9–11]. Numerical computer simulation can realize the digital simulation of UAV flight by creating 3D models, as well as simulation scenarios. Computer numerical simulation is also used in the fields of flight aircraft planning, as well as flight-simulation demonstration [12–14]. Although computer numerical simulation can significantly reduce the cost, it usually ignores the effects of the interactions between the various physical components of the actual UAV system; therefore, the simulation results usually differ from the actual situation.

The hardware-in-the-loop simulation system, which includes the simulation computer and a simplified version of the simulation hardware, is the second stage of flight simulation [15–18]. For example, KASSANDRA, a distributed architecture developed by Euroworks, allows communication between different simulation tools, and real hardware units can be seamlessly integrated with simulation units to achieve more accurate simulation [19]. A hardware-in-the-loop simulation of the UAV proportional–integral–derivative (PID) controller is proposed [20]. The controller used was a PID controller, tuned using the Ziegler–Nichols method. The implementation of the hardware-in-the-loop-simulation (HILS) can be performed after the design of control systems for UAVs is completed. These simulation systems have deterministic real-time simulation capability as well as data-acquisition capability, and when combined with computer-simulation technology, can more realistically simulate the verification of aircraft flight conditions in various environments. Hardware-in-the-loop simulation systems use computer models instead of some sensors or actuators, which have large errors compared to real sensors. Such systems cannot provide real-world visual feedback and are not suitable for the visual-navigation testing of UAVs.

A semi-physical simulation system is a system that combines the real test objects and sensors with some physical models from the simulation computer. The resulting simulation system also provides a realistic simulation of the natural environment [21–23]. Compared with the hardware-in-the-loop simulation system, it has a higher degree of simulation and is the closest indoor-simulation system to real tests. The semi-physical simulation can directly represent the parts of the flight system that cannot be accurately described by mathematical models with physical objects. The simulation of UAV flight-motion characteristics is also the focus of semi-physical simulation, and the three-axis rotary table and five-axis rotary table are commonly used to simulate the flight attitude. The S-458R-5Se infrared and laser-simulation rotary table developed in the United States can achieve 2″ accuracy in its simulated turning angle [24]. In 2018, Yonsei University in South Korea used motion-capture cameras and two pneumatic spacecraft simulators to simulate spacecraft-thrust control on a smooth aluminum surface to validate tests of autonomous navigation algorithms [25], but this system lacks *Z*-axis motion simulation and *X*- and *Y*-axes rotation. Most of the existing large rotary tables can only be used to

realize the simulation of flight attitude due to the limited number of degrees of freedom, and cannot simulate the trajectory and flight scene, which is not suitable for testing the UAV visual-navigation algorithm.

Motion-capture systems are also very common means of testing navigation algorithms on unmanned systems. For example, the OptiTrack Prime ×13 motion-capture camera has a resolution of 1.3 MP, 3D accuracy of +/−0.2 mm, and native frame rate of 240 FPS. The data-transmission latency for the motion capture-systems is typically in the millisecond range, such as 2–3 ms for systems based on the OptiTrack Prime ×13 motion capture camera. If using the EtherCAT-based semi-physical simulation platform, data-transmission latency can be reduced to microsecond range and the repeat-positioning accuracy can be further improved.

The innovations of this paper are as follows:

- Proposing a novel UAV semi-physical-simulation platform architecture that can simulate the flight attitude of a UAV in a three-dimensional space of 4 × 2 × 1.4 m realistically.
- The real-time flight-attitude simulation is realized in a real physical environment and virtual-animation space, and the flight tests of various scenes are realized by combining with the simulation-environment sandbox.
- The repeated-positioning accuracy and dynamic-positioning accuracy meet the simulation requirements of high-precision visual UAV navigation.

The rest of this paper is organized as follows: in the second section, optical-flow-field theory and its application to navigation are introduced. In the third section, a mathematical model of the simulation platform and the mechanical structure's design is presented. In the fourth section, the structure of the semi-physical simulation platform is studied and analyzed to provide the framework for the control system, and the adapted upper-computer software is also presented. In the fifth section, the performance tests of the semi-physical simulation platform, dynamic performance and optical flow are reported. In the last section, the results and limitations of this study are presented, and an outlook on future research directions is given.

## 2. Optical-Flow-Field Theory and Its Application to Navigation

The basic concept of the optical-flow method is shown in Figure 1. It was first proposed by Gibson, J.J. [26] in 1950 and is an important branch of computer vision. When the human eye looks at a moving object, the object forms constantly changing images on the retina, and these images move across the retina, creating optical flow. The ultimate goal of the optical-flow method is to calculate the object's motion information contained in two adjacent images through certain mathematical methods.

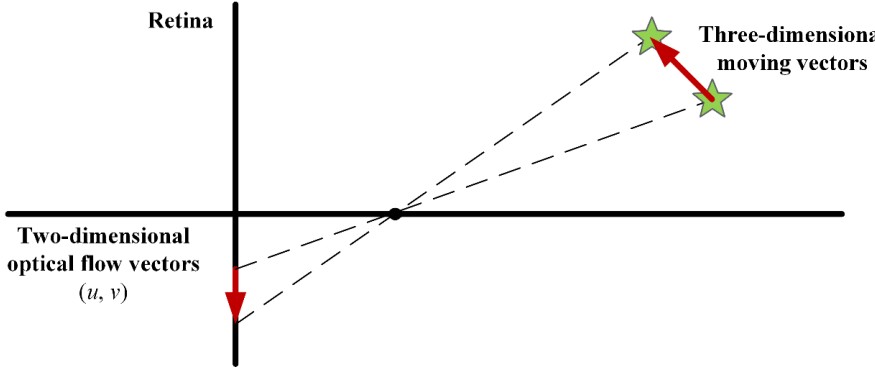

**Figure 1.** Optical-flow-field theory.

The existing method for estimating the optical flow-field assumes that the following assumptions are met: 1. Grayscale is stable and unchanged. 2. Time continuity or the movement is "small movement". The operation of the navigation system depends on the accuracy and high update rate of the navigation data, so the optical-flow algorithm for navigation should guarantee not only the accuracy, but also the computational efficiency. The Lucas–Kanade (LK) optical-flow algorithm [27] and the edge histogram-based matching algorithm are suitable for optical-flow navigation due to the fast computation speed and sparse optical-flow output. In addition to the assumption of small movements and the principle of luminance conservation, the principle of small-window-optical-flow consistency is also assumed. The assumption of optical-flow consistency is used as a basis to create a matrix of the neighborhood pixel system, through which the motion of the central pixel point in both directions is calculated. When the neighborhood pixel size is n × n, the following set of equations is obtained.

$$\begin{bmatrix} I_x & I_y \end{bmatrix} \begin{bmatrix} u \\ v \end{bmatrix} = -I_t \tag{1}$$

where $I_x$, $I_y$, and $I_t$ is the partial derivative of the grayscale of the pixel points in the image with respect to the $x$, $y$, and $t$ directions. The equation can be abbreviated as follows:

$$A\vec{n} = -\vec{b} \tag{2}$$

The least-squares method is used to solve this superdeterministic problem:

$$\begin{bmatrix} \sum I_x I_x & \sum I_x I_y \\ \sum I_x I_y & \sum I_y I_x \end{bmatrix} \begin{bmatrix} u \\ v \end{bmatrix} = - \begin{bmatrix} \sum I_x I_t \\ \sum I_y I_t \end{bmatrix} \tag{3}$$

When $A^T A$ is invertible, the value of the optical flow is obtained by solving this equation:

$$\begin{bmatrix} u \\ v \end{bmatrix} = \begin{bmatrix} \sum I_x I_x & \sum I_x I_y \\ \sum I_x I_y & \sum I_y I_x \end{bmatrix}^{-1} \begin{bmatrix} -\sum I_x I_t \\ -\sum I_y I_t \end{bmatrix} \tag{4}$$

The optical-flow algorithm based on edge-histogram matching greatly reduces the complexity of the algorithm and improves the speed of optical-flow calculation. Figure 2 shows the flow chart of the algorithm. By assuming the consistency of the optical flow, the LK optical-flow method combines the information of the neighboring pixels, eliminating the uncertainty of the optical-flow equation. However, the LK optical-flow method also has inherent shortcomings, such as the problem of large displacement and the problem of optical-flow inconsistency, which need to be improved by taking appropriate measures in the actual application. The basic steps of the large-displacement optical-flow algorithm based on edge-histogram matching are as follows.

1. Capture the image, record the continuous video frames with the camera, and save the two adjacent frames as $I_i$ and $I_{i+1}$.
2. Grayscale the two adjacent frames to obtain $I_{gray\_i}$ and $I_{gray\_(i+1)}$.
3. The grayscale image is downsampled to obtain $I_{d-sample\_i}$ and $I_{d-sample\_(i+1)}$.
4. Use the image edge-histogram-matching algorithm to obtain the coarse optical flow $(u_s, v_s)$.
5. The $i$ times of the obtained coarse optical flow are used as translations to perform pre-panning of $I_i$ and $I_{i+1}$ to obtain $I_{trans\_i}$ and $I_{trans\_(i+1)}$.
6. Solve the optical flow between $I_{trans\_i}$ and $I_{trans\_(i+1)}$ to obtain $(u_d, v_d)$, using the LK optical-flow algorithm.
7. The total optical flow $(u_f, v_f)$ is obtained by summing $(u_s, v_s)$ and $(u_d, v_d)$.

For monocular cameras, to obtain a more accurate displacement map, the distance between the camera and the captured feature point must be known. When validating flight-navigation algorithms, having more accurate distances available in real time is important for algorithm optimization.

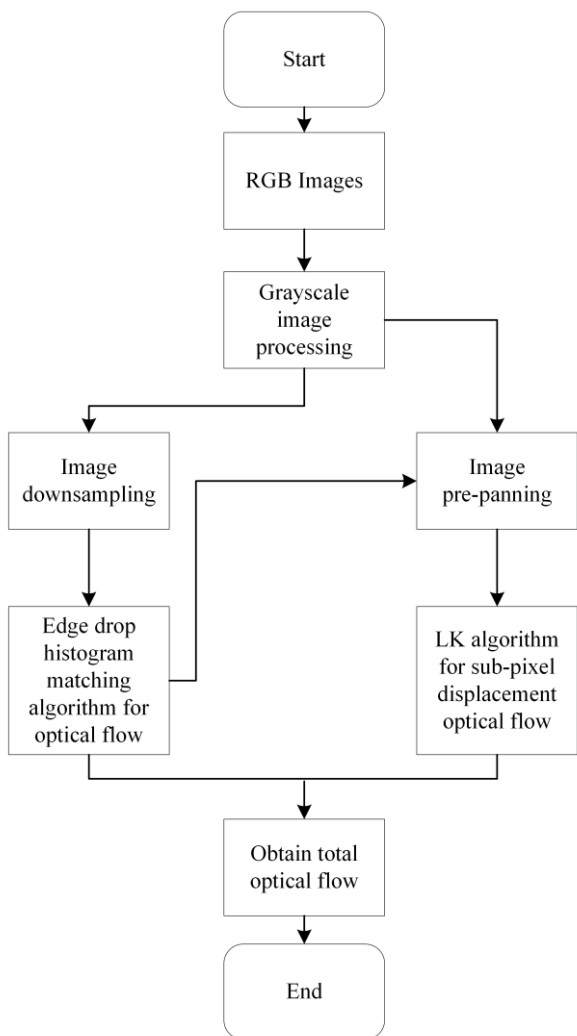

**Figure 2.** Flowchart of large-displacement optical-flow based on edge-histogram matching.

### 3. Mathematical Model of the Simulation Platform and the Mechanical Structure Design

Hardware-in-the-loop simulation and semi-physical simulation tests can usually be conducted before carrying out the real indoor or outdoor flight tests of UAVs. Using a semi-physical simulation platform for flight testing can reduce the collisions caused by uncontrollable factors during flights, avoid damaging UAVs, and improve testing efficiency. Meanwhile, the EtherCAT-based semi-physical simulation platform has lower data-transmission latency. The data-refresh cycle is less than 100 us using EtherCAT bus. The synchronization accuracy of each slave-node device can be guaranteed to be less than 1 us. It is much higher than the motion-capture system, which has a data-transmission latency in the millisecond range. In addition, the repeat positioning error of the semi-physical simulation system is usually lower than that of the motion-capture system.

This simulation platform can simulate many types of UAV, and the following is an example of a quadrotor UAV. Based on the dynamic model of the UAV and the characteristics of the optical-flow-method navigation algorithm, the UAV's semi-physical simulation platform is designed and built, and at the same time, an analysis of the dynamic performance is performed.

### 3.1. Quadrotor UAV-Dynamics Modeling

Based on the flight dynamics of the quadrotor UAV, the attitude equation of the quadrotor can be derived as follows. The inertial coordinate system (OXYZ) and quadrotor coordinate system (O'rio) are shown in Figure 3. Define the following quantities: angle of rotation $\phi$ of the roll axis; angle of rotation $\theta$ of the pitch axis; angle of rotation $\psi$ of the pitch axis. The angular velocities of the $\phi$, $\theta$, and $\psi$ are $\dot{\phi}$, $\dot{\theta}$, and $\dot{\psi}$, respectively. The angular acceleration of the $\phi$, $\theta$, and $\psi$ are $\ddot{\phi}$, $\ddot{\theta}$, and $\ddot{\psi}$, respectively. The *M1*, *M2*, *M3*, and *M4* denote the drive motors on each axis of the quadrotor. The $F_1$, $F_2$, $F_3$, and $F_4$ are the lift forces generated by the four motors. The $I_{xx}$, $I_{yy}$, and $I_{zz}$ denote the rotational inertia of each axis of the quadrotor UAV in its coordinate system, $\omega_j$ is the rotational speed of each axis motor, $\gamma$ is the rotational air drag, and $L$ is the distance from the center of motor rotation to the origin of the quadrotor coordinate system. To simplify the calculation, the inertial coordinate origin is coincident with the origin of the UAV coordinate system. The attitude equation of the quadrotor can be derived from the flight dynamics of the quadrotor UAV as follows [28]:

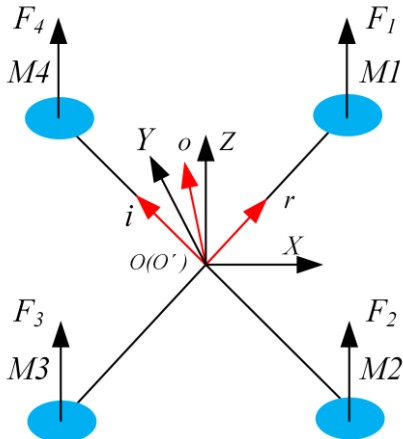

**Figure 3.** Analysis of quadrotor UAV dynamics.

$$\begin{cases} \ddot{\phi} = \dot{\theta}\dot{\psi}\left(\frac{I_{yy}-I_{zz}}{I_{xx}}\right) + \frac{L(F_4-F_2)}{I_{xx}} \\ \ddot{\theta} = \dot{\phi}\dot{\psi}\left(\frac{I_{zz}-I_{xx}}{I_{yy}}\right) + \frac{L(F_1-F_3)}{I_{yy}} \\ \ddot{\psi} = \dot{\phi}\dot{\theta}\left(\frac{I_{xx}-I_{yy}}{I_{zz}}\right) + \frac{\sum_{j=1}^{4}(-1)^j\gamma\omega_j^2}{I_{zz}} \end{cases} \quad (5)$$

The mathematical model of the drone position is as follows:

$$\begin{cases} \ddot{r} = \frac{\sum_{j=1}^{4}\rho\omega_j^2}{m}(\sin\psi\sin\phi + \cos\psi\cos\phi\sin\theta) \\ \quad - \frac{k_1}{m}\dot{r} + d_1 \\ \ddot{i} = \frac{\sum_{j=1}^{4}\rho\omega_j^2}{m}(\sin\psi\sin\theta\cos\phi - \cos\psi\sin\phi) \\ \quad - \frac{k_2}{m}\dot{i} + d_2 \\ \ddot{o} = \frac{\sum_{j=1}^{4}\rho\omega_j^2}{m}\cos\theta\cos\phi - g - \frac{k_3}{m}\dot{o} + d_3 \end{cases} \quad (6)$$

where *r*,*i*, and *o* represent the displacement on the corresponding axis in the quadrotor coordinate system, respectively, $\rho$ represents the lift coefficient, *m* is the mass of the UAV, *g* is the acceleration due to gravity, $k_1$, $k_2$, and $k_3$ are the air-drag coefficients of the UAV along the *r*, *i*, and *o* axes, $d_1$, $d_2$, and $d_3$ represent the system perturbations of the UAV along the *r*, *i*, and *o* axes. From the analysis of Equations (5) and (6), it can be observed that the attitude of the quadrotor UAV must change whenever its motion state in the three-dimensional space changes. Therefore, to simulate the flight of the UAV, a mechanical structure with six degrees of freedom in a certain space must be designed.

### 3.2. Mechanical-Structure Design

To better perform UAV testing, a three-axis rotary table was combined with a three-axis translation table in this study. The structure is capable of linear displacement in three directions and can have six degrees of freedom in a given space. Its three-dimensional structure is shown in Figure 4.

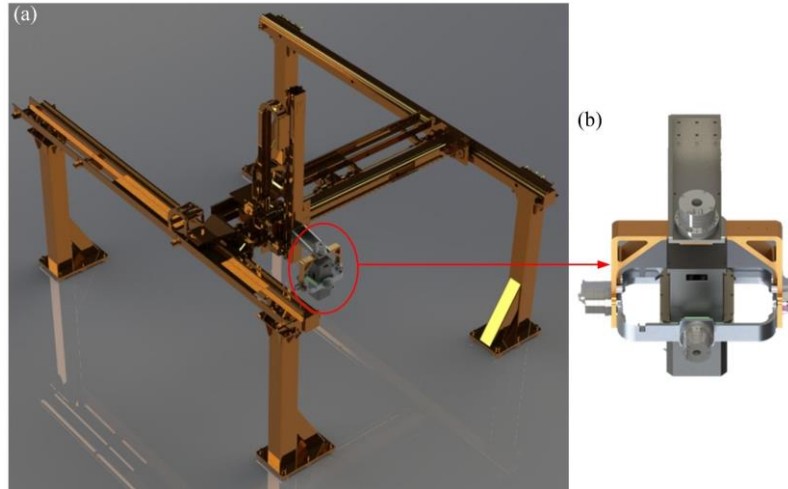

**Figure 4.** Six-axis-simulation-platform mechanical structure: (**a**) Three-axis-translation-table mechanical structure, (**b**) Three-axis-rotary-table mechanical structure.

In the simulation of quadrotor UAV flight, the end of the rotary table must support the cameras, while the rotary table is affected by the rotational inertia during operation, which imposes certain requirements on the structural strength of the rotary table.

In this study, the conventional rotary-table structure is improved to achieve a lightweight design without changing its load-bearing capacity, as shown in Figure 4b. The mechanical-performance analysis of the yaw-axis frame is shown in Figure 5. When the rotary table is suspended on the translation table, the outer frame is subjected to the maximum load. The finite-element analysis shows that the end deformation at maximum load is only $1.14 \times 10^{-7}$ mm, which meets the requirements of high-precision simulation.

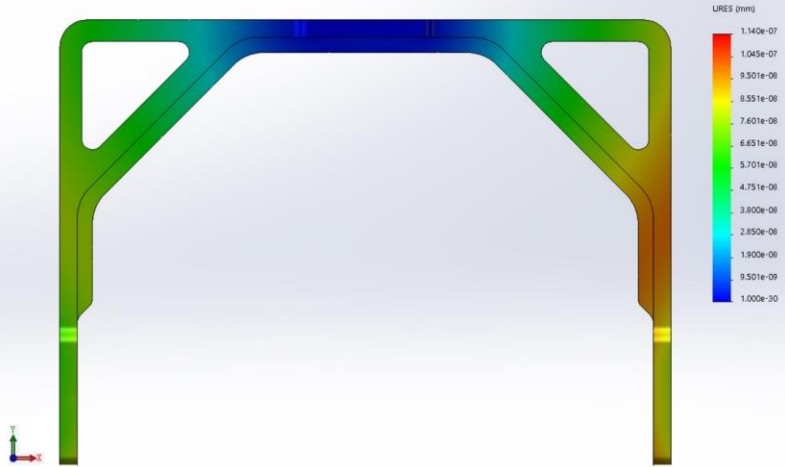

**Figure 5.** Mechanical-performance analysis of the yaw axis frame.

### 3.3. Coupled-Dynamics Model of a Six-Axis Simulation Platform

The mechanical structure of the simulation platform designed in this paper mainly consists of two subsystems: a three-axis translation table and a three-axis rotary table. The three rotating frames of the rotary table are coupled together, and their motions influence each other. The coupling mainly includes inertial coupling and dynamic coupling. Inertial coupling refers to the rotational inertia that varies within a certain range during the motion of the rotary table; kinetic coupling refers to the cross-coupling of the moments of inertia and gyroscopic effects between the frames. Therefore, the coupling between the axes must be calculated. Figure 6 shows the coordinate relationship between the three-axis translation table and the three-axis rotary table.

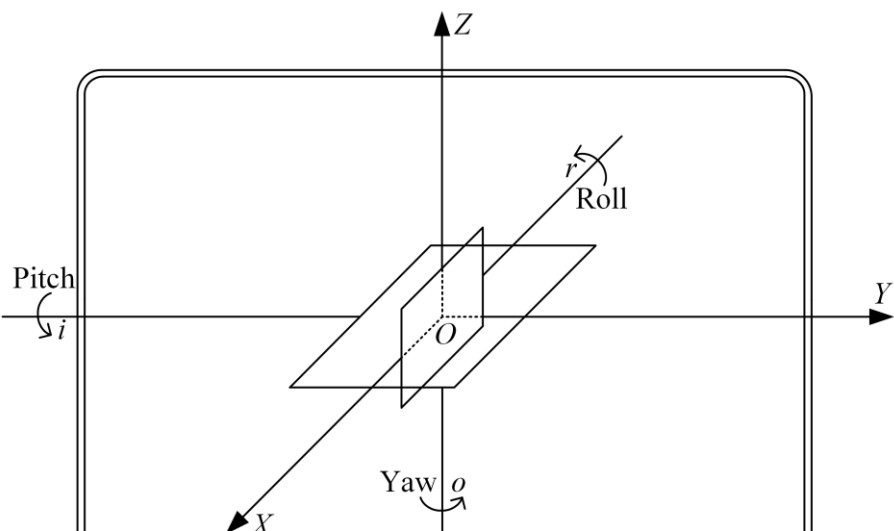

**Figure 6.** Coordinate relationship between the three-axis translation table and the three-axis rotary table.

The coordinate relationship between the three axes of the three-axis rotary table when they are not orthogonal is shown in Figure 7. The coordinate system $OXYZ$ represents the geodetic coordinate system, the coordinate system $OX_oY_oZ_o$ represents the yaw-axis-frame coordinate system, the coordinate system $OX_iY_iZ_i$ represents the pitch-axis-frame coordinate system, and the coordinate system $OX_rY_rZ_r$ represents the roll-axis-frame coordinate system. When the three axes are orthogonal to each other, the yaw axis (including the pitch and roll axis frames) is rotated counterclockwise by one angle $\gamma$ about its axis of rotation $OZ_o(OZ)$. The pitch axis (including the roll-axis frame) is rotated counterclockwise by an angle $\beta$ around its axis of rotation $OY_i$. The roll axis rotates counterclockwise by an angle $\alpha$ around its rotation axis $OX_r$. This results in the coordinate-relationship diagram in Figure 7. The angular velocities of the roll, pitch, and yaw axes are $\dot{\alpha}$, $\dot{\beta}$, and $\dot{\gamma}$, respectively. The angular of the roll-, pitch-, and yaw-axes accelerations are $\ddot{\alpha}$, $\ddot{\beta}$, and $\ddot{\gamma}$, respectively. The $J_N$ represents the rotational inertia of the corresponding axis, where $N = X$, $Y$, and $Z$. The $J_{M_n}$ represents the rotational inertia of the corresponding axis, where $M = X$, $Y$, and $Z$, $n = i$ and $r$.

The rotational inertia $J_{Y_{ri}}$ of the roll-axis frame with respect to the axis $OY_i$ can be calculated as follows:

$$J_{Y_{ri}} = J_{Y_r} \cos^2 \alpha + J_{Z_r} \sin^2 \alpha \tag{7}$$

The rotational inertia $J_{Z_{ro}}$ of the roll-axis frame with respect to the axis $OZ_o$ can be calculated as follows:

$$J_{Z_{ro}} = J_{X_r} \sin^2 \beta + J_{Y_r} \sin^2 \alpha \cos^2 \beta + J_{Z_r} \cos^2 \alpha \cos^2 \beta \tag{8}$$

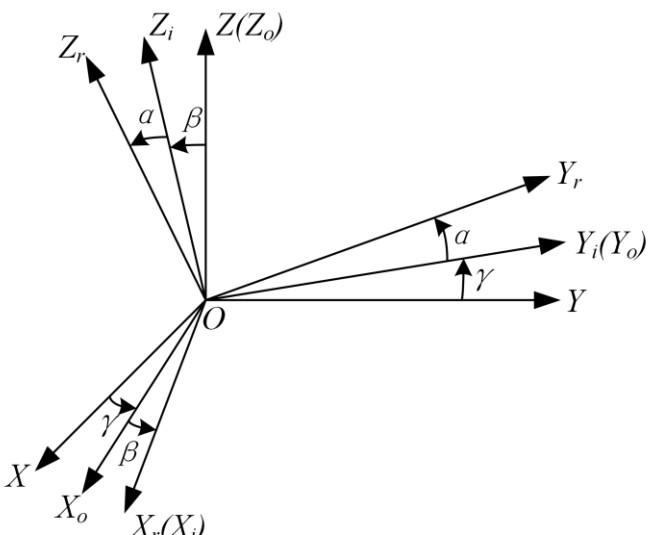

**Figure 7.** Coordinate transformation of three-axis rotary table.

The rotational inertia $J_{Z_{io}}$ of the roll-axis frame with respect to the axis $OZ_o$ can be calculated as follows:

$$J_{Z_{io}} = J_{X_i} \sin^2 \beta + J_{Z_i} \cos^2 \beta - J_{X_i Z_i} \sin \beta \cos \beta - J_{Z_i X_i} \sin \beta \cos \beta + J_{X_i} \sin^2 \beta \qquad (9)$$

The rotational inertia $J_{OX_r}$ of the pitch-axis frame with respect to the axis $OX_r$ can be calculated as follows:

$$J_{OX_r} = J_{X_r} \qquad (10)$$

The rotational inertia $J_{OY_i}$ of the pitch-axis frame, including the roll-axis frame, with respect to the pitch axis $OY_i$ can be calculated as follows:

$$J_{OY_i} = J_{Y_{ri}} + J_{Y_i} = J_{Y_i} + J_{Y_r} \cos^2 \alpha + J_{Z_r} \sin^2 \alpha \qquad (11)$$

From Equation (11), the inertia $J_{OZ_o}$ of the yaw frame, including the roll frame and the pitch frame, with respect to the yaw axis $OZ_o$ can be deduced as follows:

$$
\begin{aligned}
J_{OZ_o} &= J_{Z_o} + J_{Z_{io}} + J_{Z_{ro}} = J_{Z_o} + J_{X_i} \sin^2 \beta + J_{Z_i} \cos^2 \beta - J_{X_i Z_i} \sin \beta \cos \beta \\
&\quad - J_{Z_i X_i} \sin \beta \cos \beta + J_{X_i} \sin^2 \beta + J_{X_r} \sin^2 \beta + J_{Y_r} \sin^2 \alpha \cos^2 \beta + J_{Z_r} \cos^2 \alpha \sin^2 \beta
\end{aligned} \qquad (12)
$$

The value of $J_{OZ_o}$ can be obtained by substituting the approximate value of moment of inertia of the three axes in Table 1 with respect to each axis of rotation into Equation (12). To facilitate the calculation, the motion of each rigid body is solved separately, after which the mutual influence is calculated. The angular-velocity vectors of the three rigid bodies are defined as follows. $\omega_r = \begin{pmatrix} \omega_{X_r} & \omega_{Y_r} & \omega_{Z_r} \end{pmatrix}^{\mathrm{T}}$ is the angular-velocity vector of the roll axis with respect to the pitch axis. $\omega_i = \begin{pmatrix} \omega_{X_i} & \omega_{Y_i} & \omega_{Z_i} \end{pmatrix}^{\mathrm{T}}$ is the angular-velocity vector of the pitch axis with respect to the yaw axis. $\omega_o = \begin{pmatrix} \omega_{X_o} & \omega_{Y_o} & \omega_{Z_o} \end{pmatrix}^{\mathrm{T}}$ is the angular-velocity vector of the yaw axis with respect to the geodetic coordinate system. According to the Gothic rotation theorem [29], we obtain:

$$H = H_X i + H_Y j + H_Z k = J_X \omega_X i + J_Y \omega_Y j + J_Z \omega_Z k \qquad (13)$$

where $H$ is the moment of momentum of the rigid body, $H_x$, $H_y$, and $H_z$ represent the moment of momentum on the $x$, $y$, and $z$ axes of the rigid body, and $i$, $j$, and $k$ are the unit vectors on the $X$, $Y$, and $Z$ axes.

Let $M = \begin{pmatrix} M_X & M_Y & M_Z \end{pmatrix}^{\mathrm{T}}$ be the moment to which the rigid body is subjected. According to the momentum-moment theorem [29] we can obtain:

$$\frac{dH}{dt} = M \tag{14}$$

The Eulerian=dynamics equation for a rigid body is obtained by the union of Equations (13) and (14).

$$\begin{cases} J_X \frac{d\omega_X}{dt} + (J_Z - J_Y)\omega_Y\omega_Z = M_X \\ J_Y \frac{d\omega_Y}{dt} + (J_X - J_Z)\omega_X\omega_Z = M_Y \\ J_Z \frac{d\omega_Z}{dt} + (J_Y - J_X)\omega_X\omega_Y = M_Z \end{cases} \tag{15}$$

The final result is as follows:

$$M_{X_r} = 0.6512\ddot{\alpha} - 0.6512\ddot{\gamma}\sin\beta - 0.6512\dot{\beta}\dot{\gamma}\cos\beta \tag{16}$$

$$M_{Y_i} = 0.8140\ddot{\beta} + 0.6512\dot{\alpha}\dot{\gamma}\cos\beta + 0.1534\dot{\gamma}^2\cos\beta\sin\beta \tag{17}$$

$$\begin{aligned} M_{Z_o} &= 1.0389\ddot{\gamma} + 0.4978\sin^2\beta + 0.6395\sin(\alpha + \beta) \\ &\quad - 0.6495\ddot{\alpha}\sin\beta - 0.6495\dot{\alpha}\dot{\beta}\cos\beta \end{aligned} \tag{18}$$

**Table 1.** Approximate moments of inertia of the three axes relative to each axis of rotation.

| Rotational Inertia J/(kg.mm$^2$) | *X*-axis | *Y*-axis | *Z*-axis |
|---|---|---|---|
| Roll axis | 0.6512 | 0.6495 | 0.6295 |
| Pitch axis | 0.1657 | 0.1635 | 0.3191 |
| Yaw axis | / | / | 0.7198 |

### 3.4. AC-Servo-Motor Mathematical Model

The AC servo system has a high torque ratio and can achieve both rapid start-up and deceleration. The motor chosen in this paper is a permanent-magnet AC servo motor, and the torque equation of the motor $T_d$ is calculated according to [28]:

$$T_d = P_m\left[\psi^r i_q^s + (L_d - L_q)i_d^s i_q^s\right] \tag{19}$$

where $P_m$ is the motor power, $\psi^r$ is the coupled magnetic chain of the rotor magnets on the stator, $L_d$ and $L_q$ are the direct- and alternating-axis main inductance of the permanent magnet synchronous motor, and $i_q^s$ and $i_d^s$ are the alternating and direct axis components of the stator-current vector.

To simplify the control system, take $i_d^s = 0$ and $i_q^s = i^s$. The torque equation of the permanent-magnet AC motor is transformed into:

$$T_d = P_m\psi^r i^s \tag{20}$$

Since $P_m\psi^r$ is the motor constant, the torque equation is simplified as:

$$T_d = ki^s \tag{21}$$

where $k$ is the torque constant of the AC servo motor. An examination of the motor-product manual shows that the torque constants of the roll and pitch axes are 4 kg·fm/A, and the torque constant of the yaw axis is 4.2 kg·fm/A. The AC motor can be simplified to a DC-motor model to realize the decoupling of the control parameters of the three-phase permanent magnet synchronous motor and achieve the purpose of vector control. By coupling Equations (16)–(18) and (21) and neglecting the minimal quantities, we obtain:

$$\ddot{\alpha} = 6.14I_r + 2.64I_o\sin\beta + \dot{\beta}\dot{\gamma}\cos\beta \tag{22}$$

$$\ddot{\beta} = 4.9 I_i - 0.8 \dot{\alpha} \dot{\beta} \cos\beta - 0.1884 \dot{\gamma}^2 \cos\beta \sin\beta \tag{23}$$

$$\ddot{\gamma} = 4.05 I_0 - 0.6156 \sin(\alpha + \beta) + 3.84 I_r \sin\beta + 0.6252 \dot{\alpha} \dot{\beta} \cos\beta \tag{24}$$

where $I_r$, $I_i$, and $I_o$ are the currents along the $r$, $i$, and $o$ axes. Assuming that:

$$\begin{cases} x_1 = \alpha \\ x_2 = \dot{\alpha} \\ x_3 = \beta \\ x_4 = \dot{\beta} \\ x_5 = \gamma \\ x_6 = \dot{\gamma} \end{cases} \tag{25}$$

The dynamic system of the rotary table can then be converted into the following form:

$$\sum : \begin{bmatrix} \dot{x}_1 \\ \dot{x}_2 \\ \dot{x}_3 \\ \dot{x}_4 \\ \dot{x}_5 \\ \dot{x}_6 \end{bmatrix} = \begin{bmatrix} x_2 \\ \ddot{\alpha} \\ x_4 \\ \ddot{\beta} \\ x_6 \\ \ddot{\gamma} \end{bmatrix} \tag{26}$$

$$u = \begin{bmatrix} I_o \\ I_i \\ I_r \end{bmatrix} \tag{27}$$

$$\begin{bmatrix} y_1 \\ y_2 \\ y_3 \end{bmatrix} = \begin{bmatrix} x_1 \\ x_3 \\ x_5 \end{bmatrix} \tag{28}$$

It can be seen that the three-axis rotary table is a non-linear system with three inputs and three outputs, and the individual rotating axes are coupled together. Therefore, to improve the control accuracy, the system must be decoupled for calculation.

The system can be decoupled using state feedback and dynamic-feedback compensation. For Equation (26), let:

$$\begin{cases} \varphi_1 = \frac{\dot{x}_2}{6.24} \\ \varphi_2 = \frac{\dot{x}_4}{4.9} \\ \varphi_3 = \frac{\dot{x}_6}{4.05} \end{cases} \tag{29}$$

where $\varphi_1$, $\varphi_2$, and $\varphi_3$ are the current corrections for the $r$, $i$, and $o$ axes. Next, there are:

$$I_r = \varphi_1 - \frac{1}{6.14}(2.64 I_o \sin x_3 + x_4 x_6 \cos x_3) \tag{30}$$

$$I_i = \varphi_2 - \frac{1}{4.9}(-0.8 x_2 x_4 \cos x_3 - 0.1884 x_6^2 \cos x_3 \sin x_3) \tag{31}$$

$$I_0 = \varphi_3 - \frac{1}{4.05}(-0.6156 \sin(x_1 + x_3) + 3.84 I_r \sin x_3 + 0.6252 x_2 x_4 \cos x_3) \tag{32}$$

If Equations (30)–(32) are connected in series as a dynamic-compensation and state-feedback decoupling network before Equation (26), the system can be reduced as follows:

$$\sum : \begin{bmatrix} \dot{x}_1 \\ \dot{x}_2 \\ \dot{x}_3 \\ \dot{x}_4 \\ \dot{x}_5 \\ \dot{x}_6 \end{bmatrix} = \begin{bmatrix} x_2 \\ 6.24\varphi_1 \\ x_4 \\ 4.9\varphi_2 \\ x_6 \\ 4.05\varphi_3 \end{bmatrix} \tag{33}$$

$$u = \begin{bmatrix} \varphi_1 \\ \varphi_2 \\ \varphi_3 \end{bmatrix} \tag{34}$$

$$\begin{bmatrix} y_1 \\ y_2 \\ y_3 \end{bmatrix} = \begin{bmatrix} x_1 \\ x_3 \\ x_5 \end{bmatrix} \tag{35}$$

In this case, the system is a single-input, single-output system with inputs $\varphi_1$, $\varphi_2$, and $\varphi_3$ and outputs $y_1$, $y_2$, and $y_3$. The decoupling network is connected in series with the control network, as shown in Figure 8.

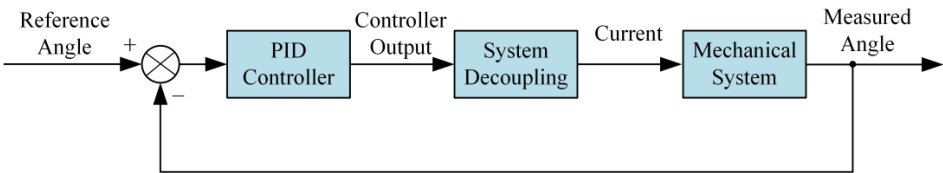

**Figure 8.** Decoupled control schemes.

## 4. Control-System Design

The most important part of the quadrotor UAV visual-navigation semi-physical simulation platform's design is the servo control system. The control system must have a fast response, high accuracy and good robustness. For this reason, the design of a real-time control system based on the EtherCAT bus was chosen.

### 4.1. Overall Structure

This system is a multi-axis synchronous and cooperative system that can be used to simulate a UAV flight. As shown in Figure 9, the control system consists of a workstation, a Trio PC-MACT EtherCAT bus controller, and several servo motors. The workstation is connected to the TRIO PC-MCAT motion controller via Ethernet. The motion controller is connected to I/O modules via EtherCAT for simultaneous connection to six servo-motor drives. This allows six motors to rotate simultaneously to achieve a simulated flight attitude. Servo motors use Panasonic and Harmonic AC servo motors. They have a unique miniaturized design and a hollow-hole structure. The through-hole in the center of the actuator can pass through wiring, piping, lasers, and so on. This simplifies the overall structure of the mechanism.

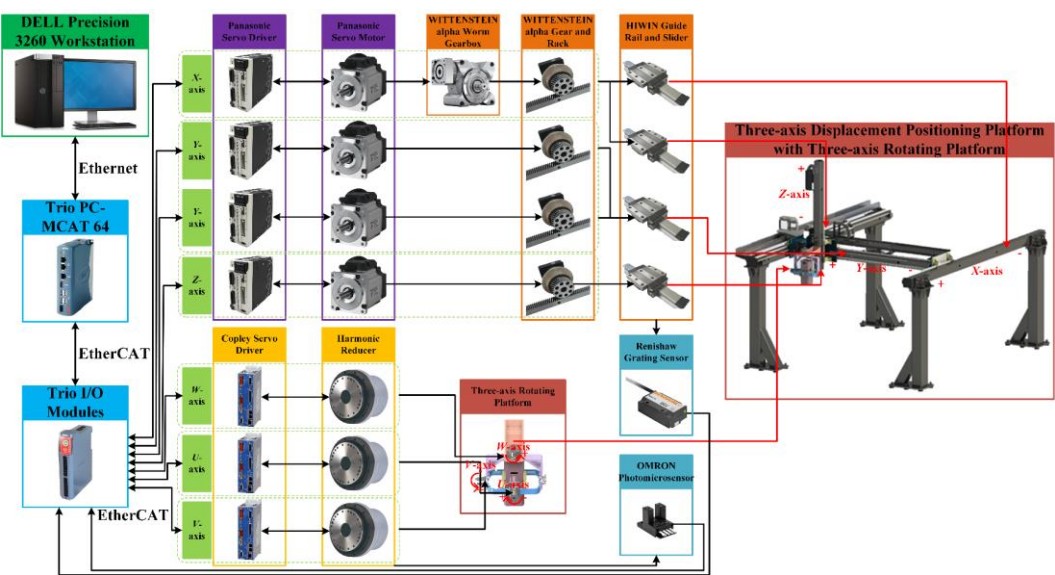

**Figure 9.** Overall architecture of the control system.

For the control module of the quadrotor UAV simulation system, this system supplies power to each axis independently and develops overload and leakage protection. The power-supply filter is also installed to enhance the anti-interference capability of the servo-motor driver and improve the control stability of the system. The electrical connection is shown in Figure 10.

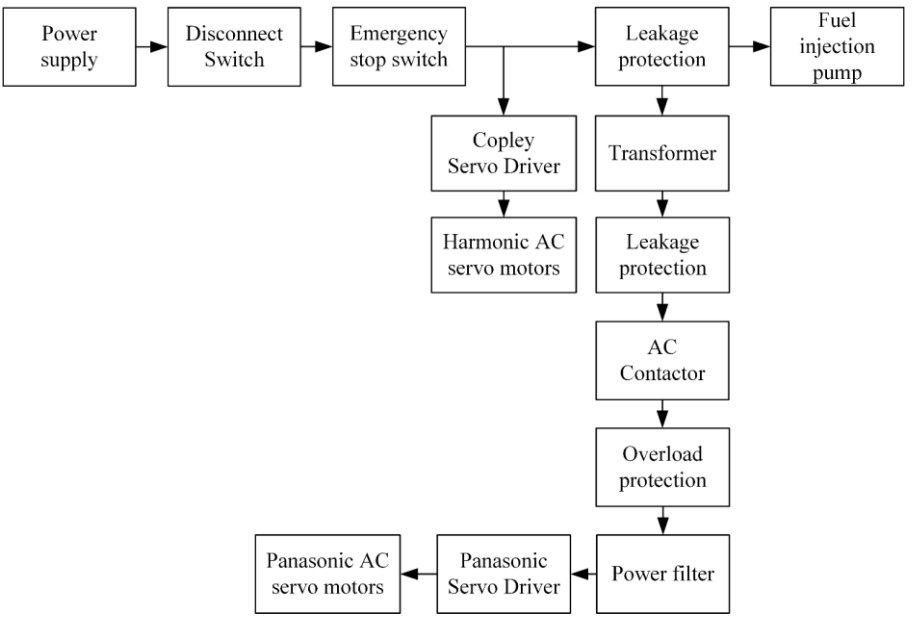

**Figure 10.** Schematic diagram of electrical connections.

*4.2. Maximum Motor Load*

As shown in Table 2, the load on the Z-axis is the largest, reaching 2918.7 N, where the speed ratio of the speed reducer is 7. According to the calculation, the maximum torque of the motor is 8.85 N·m, and the maximum speed is 2519.98 r/min. Taking into account the safety-factor ratio, the system chooses the Panasonic MHMF402L1 motor. The maximum output torque of the motor is 57.3 N·m, and the torque-utilization ratio is 33%. Its maximum speed is 3000 r/min and the speed-utilization rate is 84%. It can provide stable power for the Z-axis. Similarly, Panasonic MHNF152L1 and MHMF03L1 servo motors were selected for the X and Y axes, respectively.

**Table 2.** Main parameters of the three-axis translation table.

| Design Parameters | *X*-axis | *Y*-axis | *Z*-axis |
|:---:|:---:|:---:|:---:|
| Maximum speed/(mm/s) | 1000 | 1000 | 1000 |
| Maximum acceleration/(mm/s$^2$) | 1000 | 1000 | 1000 |
| Linear load/N | 1643.5 | 775.1 | 2918.7 |

From Table 3, it can be calculated that the angular-velocity-design parameter is 20°/s. Converting this to radians, the angular velocity is:

$$\omega = \frac{20 \times \pi}{180} = 0.3489 \ (\text{rad/s}) \tag{36}$$

**Table 3.** Main parameters of the three-axis rotary table.

| Design Parameters | *W*-axis | *U*-axis | *V*-axis |
|---|---|---|---|
| Maximum speed/(°/s) | 20 | 20 | 20 |
| Maximum acceleration/(°/s$^2$) | 20 | 20 | 20 |
| Linear load/(kg·mm$^2$) | 1.04 | 0.66 | 0.81 |

If the acceleration curve is based on the trapezoidal acceleration and deceleration curve, the acceleration time should not exceed 0.1 s for the fast-moving mechanical structure. Therefore, the acceleration time $t_{ac}$ is set to 0.05 s, from 0 to 20°/s. The acceleration $\omega_{ac}$ is:

$$\omega_{ac} = \frac{\omega}{t_{ac}} = 6.978 \left( \text{rad}/s^2 \right) \tag{37}$$

The acceleration torque $T_{ac}$ of the rotary table is:

$$T_{ac} = \text{J}_{xx} \times \omega_{ac} = 1.043 \times 6.978 = 7.278 (\text{N} \cdot \text{m}) \tag{38}$$

where $\text{J}_{xx}$ represents main rotational inertia at center. Considering the load torque of the rotary table $T_L = 0$, the friction torque of the bearing when the rotary table rotates is small and can be neglected. If the mechanical-drive efficiency of the motor $\eta = 0.9$, the torque $T_M$ of the motor is:

$$T_M = \frac{T_{ac} + T_L}{\eta} = 8.086 (\text{N} \cdot \text{m}) \tag{39}$$

For motors with fast motion and frequent starts and stops, it is also necessary to calibrate the equivalent torque. Consider the most extreme case, where the rotary table is rotated from the positive limit to the negative limit and back to the positive limit, as the operating cycle. Based on the load torque $T_L = 0$ of the rotary table, it is obvious that the average torque is at its maximum when the uniform motion time is 0. Since the load torque is 0, it is estimated that the equivalent torque is still the acceleration torque. The equivalent torque $T_{rms}$ can be expressed as:

$$T_{rms} = \sqrt{\frac{T_{ac}^2 \times t_{ac} + T_{ac}^2 \times t_{ac} + T_{ac}^2 \times t_{ac} + T_{ac}^2 \times t_{ac}}{4 \times t_{ac}}} = T_{ac} \tag{40}$$

For the functional requirements of rotation, the Harmonic-SHA25A101S-integrated AC servo actuator was selected for the system. Since the harmonic reducer with a reduction ratio of 101 is used, the motor has a large equivalent rotational inertia. The motor can provide a torque of approximately 70 N·m under continuous operation or a maximum torque of nearly 200 N·m for short overloads; therefore, the motor can meet the requirements for rotary-table torque with a large safety margin. Since the W-axis is the most heavily loaded of the three axes, the other two axes can also meet the performance requirements with the same type of motor.

*4.3. System Construction*

As can be seen from the control-system architecture, the synchronization, real-time, and accuracy requirements for the operation of the individual axis motors are extremely high. The EtherCAT bus-control technology is the fastest industrial Ethernet technology, with a data transmission speed of 100 M bit/s. The data frames are processed in real time. With a data-refresh period of less than 100 us and a high-precision distributed clock, the synchronization accuracy of the slave-node devices is less than 1 μs.

Therefore, the Trio EtherCAT controller PC-MCAT was selected as the motion controller. The three-axis rotary table uses Copley servo drivers, based on EtherCAT communication, and the A/D converter has a very high resolution to ensure the best current-loop performance. The three-axis translation table uses Panasonic MDDLNT55BF servo drivers with associated servo motors. At the same time, a Renishaw HK-0400-0002 encoder is mounted on the linear guide to provide feedback on the position values. The real system is shown in Figure 11. A special simulation-environment sandbox is set up in the space below the simulation system for simulating various scenes with aerial textures, as shown in Figure 11c.

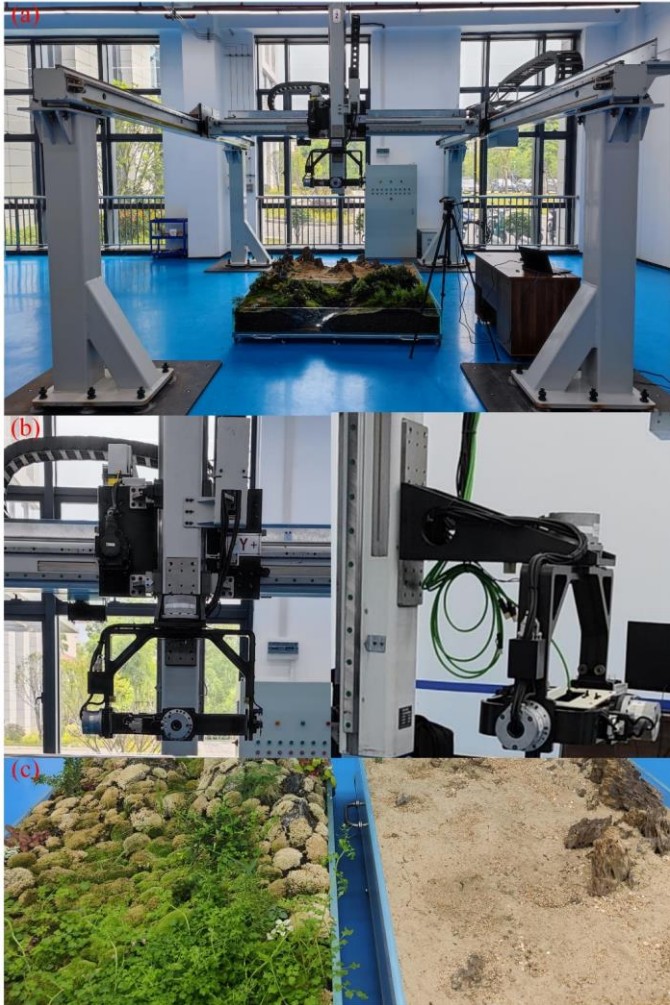

**Figure 11.** Real system pictures of the simulation system: (**a**) front view, (**b**) rotary table, (**c**) simulation-environment sandbox.

### 4.4. Control-System-Software Architecture

As shown in Figure 12, the platform-application software is divided into a control module and a flight-simulation module. The control module is mainly based on the underlying control commands of the TRIO PC-MACT motion-control card, which is based on VS2017 and Qt to achieve some basic motion-control functions. The UAV's attitude information is converted to TRIO BASIC language and executed. The flight-simulation module converts the information from the simulation platform back to each axis into an animated flight simulation of the UAV.

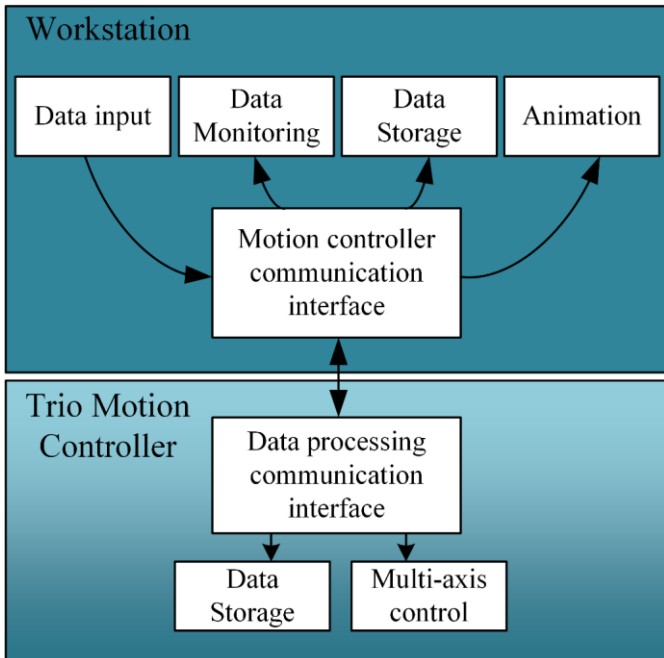

**Figure 12.** Software architecture.

As can be seen from Figure 13, based on Open Inventor, the animation module of the system converts the information of the simulation platform into the flight attitude of the UAV and the space position of the simulated scene. The trajectory of the flight is also displayed. The flowchart of the program running on the workstation is shown in Figure 14. It uses the idea of multi-threaded parallelism by running the data display, flight-animation simulation, attitude calculation, and command conversion in the opened sub-threads, respectively. Each thread communicates with others through Qt's slot signals to ensure the smooth operation of the software.

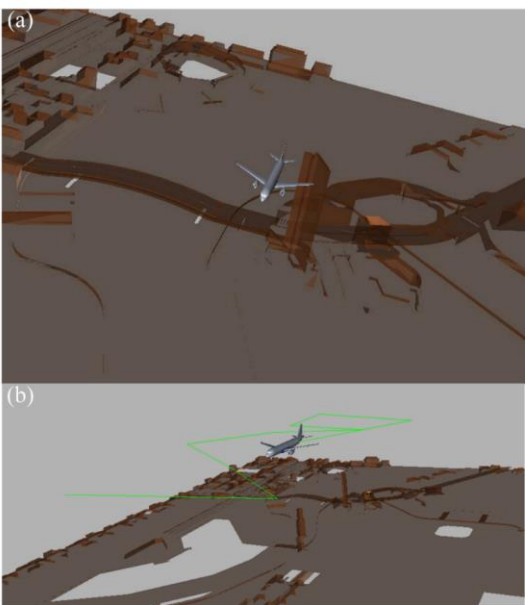

**Figure 13.** Real-time animation simulation: (**a**) Animated simulation of the flight in the city, (**b**) flight-trajectory display.

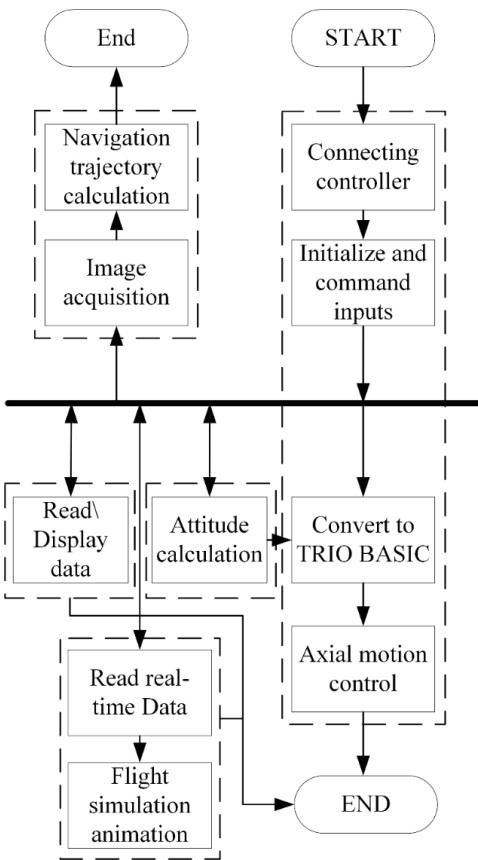

**Figure 14.** Software flowchart.

## 5. Experimental Results and Discussion

In this chapter, the repeat-positioning accuracy, cumulative-positioning accuracy, and dynamic performance of the simulation platform are tested and verified.

### 5.1. Steady-State Performance Test

To test the steady-state performance of the simulation platform, a laser interferometer was used in this study to test the linear-axis running range ($XYZ$), repeat positioning accuracy, cumulative positioning accuracy, linear-axis running speed, and linear-axis running acceleration of the three-axis translation table. In addition, the rotary-axis running range ($UVW$), repetitive positioning accuracy, rotary-axis running speed, and rotary-axis running acceleration of the three-axis rotary table were also tested.

The test method for the accuracy of the three-axis translation table involved cyclic and continuous measurement in the positive and negative directions of the $X$-, $Y$-, and $Z$-axes. Three convergences were made in each direction for each target position $P_i$. The actual arrival position was measured with a laser interferometer and the position deviation was calculated. For the cumulative error, the initial target position was determined by the motion on one axis and moving the moving part 1000 m. The moving part returned to the initial target position. The actual arrival position was measured with the laser interferometer and the position deviation was calculated as the cumulative positioning accuracy.

The test method for the accuracy of the three-axis rotary table involved the use of a laser interferometer and a supporting indexing table to calibrate the rotary axis. The indexing table was installed at the center of the rotary axis. The indexing table and the rotation center of the rotary axis were adjusted by applying the table method, so that the radial circular run-out value did not exceed 0.02 mm. The angle reflector was installed on the indexing table, and the angle reflector was made surface perpendicular to the laser beam. The angular interferometer was installed in the optical path so that it was parallel

to the angular reflector boarding, the laser head was panned so that the laser beam was collimated. The rotation speed of the control system was set and the target position, the overtravel amount, the stopping time at the target point, and the number of cycles were determine. Continuous measurements were cycled in the positive and negative directions of the *U*, *V*, and *W* axes. Three convergences were performed in each direction for each target position $P_i$. The actual arrival position was measured with the laser interferometer and the position deviation was calculated. In the tests of the cumulative positioning accuracy, the corresponding axis was moved continuously for a distance of 1000 m or more to determine this value.

The *X*-axis was repeatedly measured three times according to the specified test method, and the test results are shown in Figure 15a. The repeat-positioning accuracy was 0.033 mm, and the reverse difference was 0.11 mm. The cumulative-positioning accuracy was 0.02 mm/1000 m after moving the *X*-axis for 1000 m. As the corresponding axes needed to be made to actually move for 1000 m or more, this test was time consuming, usually between half a day and a day. The *Y*-axis was measured repeatedly three times according to the specified test method, and the test results are shown in Figure 15b. The repeat positioning accuracy is 0.012 mm, and the reverse difference is 0.11 mm. The *Z*-axis was measured repeatedly three times according to the specified test method, and the test results are shown in Figure 15c. The repeat-positioning accuracy is 0.004 mm, and the reverse difference is 0.007 mm.

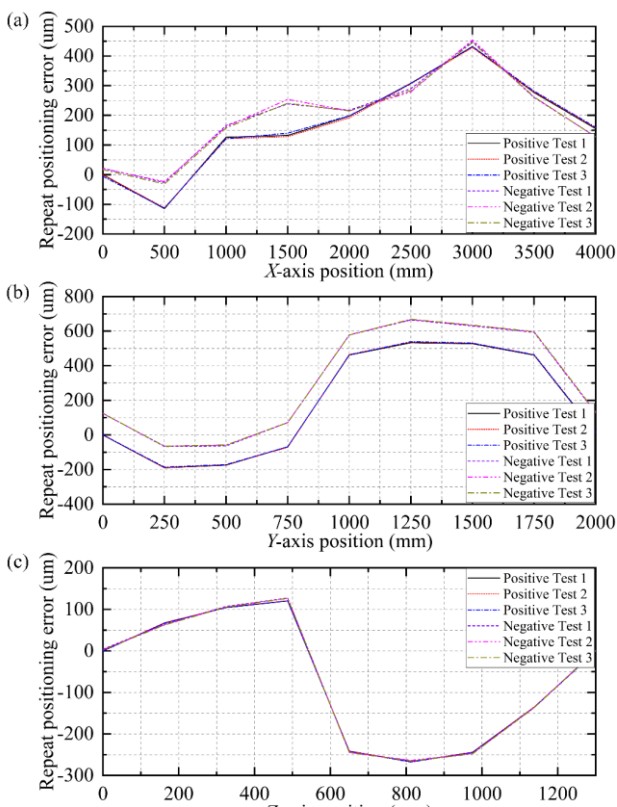

**Figure 15.** Repeat-positioning-error test results: (**a**) *X*-axis, (**b**) *Y*-axis, (**c**) *Z*-axis.

According to the specified test method of repeating the measurement of the *U*-axis three times, the test results are shown in Figure 16a; the repeat-positioning accuracy was 0.002° and the reverse difference was 0.001°. According to the established test method of repeating the measurement of the *V*-axis three times, the test results are shown in Figure 16b; the repeat positioning accuracy was 0.002° and the reverse difference was 0.001°. According to the specified test method of repeating the measurement of the *W*-axis three times, the test results are shown in Figure 16c; the repeat-positioning accuracy was

0.006° and the reverse difference was 0.001°. The measurement results are shown in the following table. Table 4 shows the steady-state-performance test results of the three-axis translation table and the three-axis rotary table.

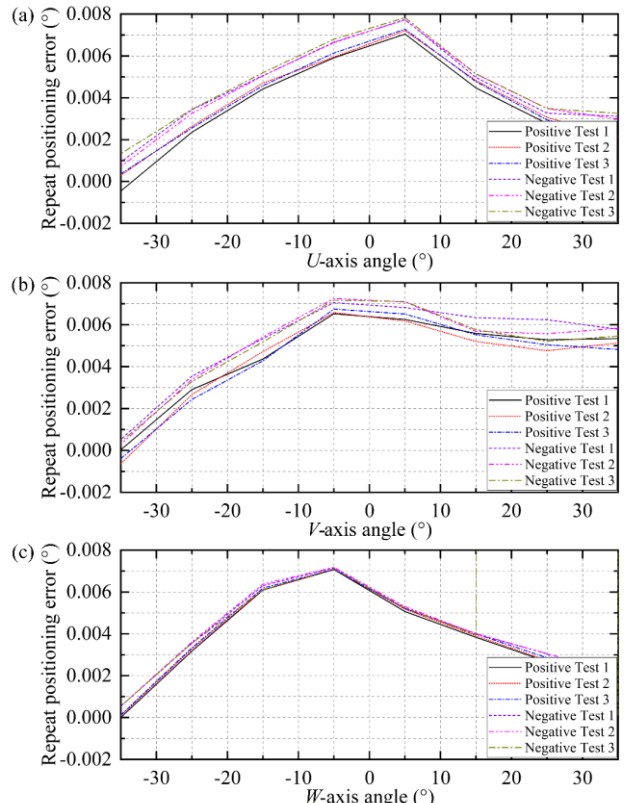

**Figure 16.** Repeat-positioning-error-test results: (**a**) *U*-axis, (**b**) *V*-axis, (**c**) *W*-axis.

**Table 4.** Steady-state performance test results.

| Table | Tests | Test Results | |
|---|---|---|---|
| three-axis translation table | Operating range (*XYZ*) | 4000 × 2000 × 1400 mm | |
| | Repeat-positioning accuracy | *X* | 0.033 mm |
| | | *Y* | 0.012 mm |
| | | *Z* | 0.004 mm |
| | Dynamic-positioning accuracy | *X* | 0.4 mm |
| | | *Y* | 0.2 mm |
| | | *Z* | 0.1 mm |
| | Cumulative-positioning accuracy | 0.02 mm/1000 m | |
| | Maximum speed | 1000 mm/s | |
| | Maximum acceleration speed | 1000 mm/s$^2$ | |
| three-axis rotary table | Operating range (*UVW*) | ±40° | |
| | Repeat-positioning accuracy | *U* | 0.002° |
| | | *V* | 0.002° |
| | | *W* | 0.006° |
| | Dynamic-positioning accuracy | *U* | 0.03° |
| | | *V* | 0.04° |
| | | *W* | 0.04° |
| | Maximum speed | 20°/s | |
| | Maximum acceleration speed | 20°/s$^2$ | |

### 5.2. Dynamic Performance Test

As shown in Figures 17 and 18, a simulated flight trajectory was selected, converted into a TRIO BASIC command and then read to obtain the actual trajectory map. After measurement and calculation, the dynamic error of the three-axis translation table was 0.4 mm.

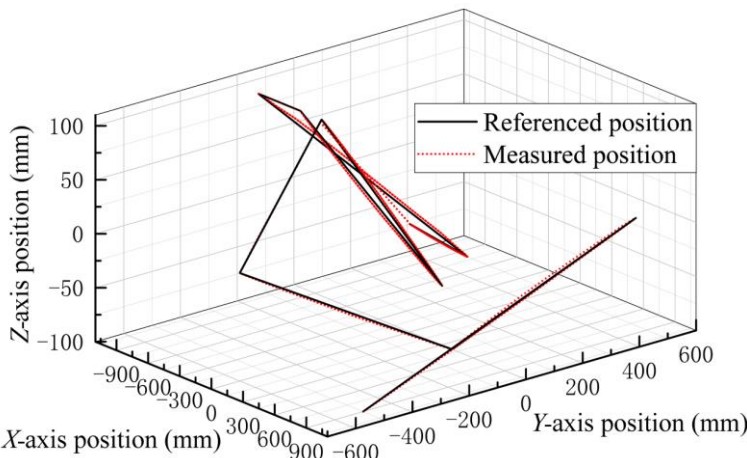

**Figure 17.** Random-curve dynamic tracking of a three-axis translation table.

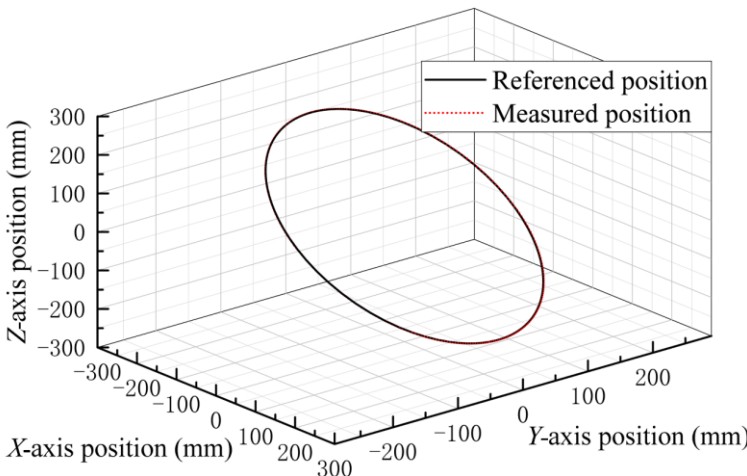

**Figure 18.** Circular-curve dynamic tracking of a three-axis translation table.

To represent the dynamic-rotation-angle error of the three-axis rotary table more intuitively, the angles of the three axes are represented by three-dimensional spatial coordinates in this paper, as shown in Figure 19. After measurement and calculation, the dynamic-rotation-angle error of the three-axis rotary table was controlled within 0.04°. Fold lines and spatial arcs were selected as the simulated flight trajectories for the test. The flight trajectory was transformed into control information, and the dynamic error was obtained by comparing the actual trajectory map with the given trajectory map after simulating the flight by the three-axis translation stage.

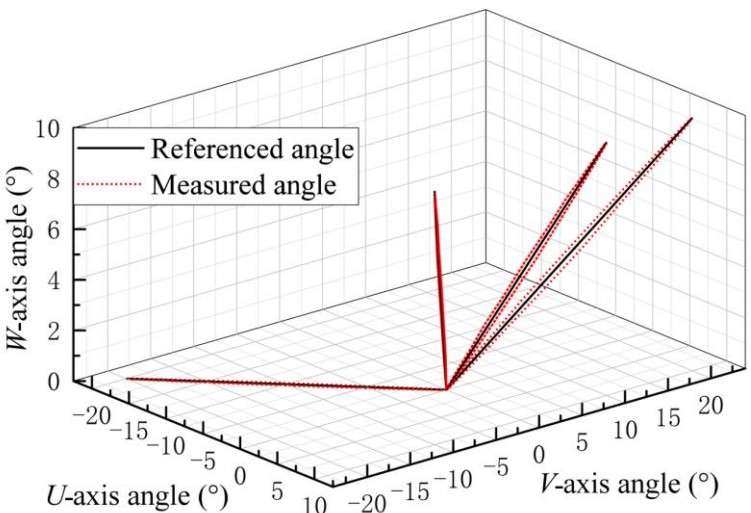

**Figure 19.** Dynamic-trajectory tracking of a three-axis rotary table.

*5.3. Optical-Flow Test*

The outdoor flight environment was scaled down and the scenario was replicated in the indoor simulation. Next, the quadrotor UAV simulation system performed the navigation test according to the predetermined trajectory. The end of the simulation system was equipped with an optical detection system, which transmitted the collected optical information to the host computer and calculated the speed using the optical-flow-navigation algorithm. Finally, the trajectory was fitted according to the velocity calculated by the optical flow. Figure 20 shows the LK optical-flow-calculation results after pretreatment during the indoor flight tests, and its feature points match accurately. The results demonstrate the effectiveness of the simulation platform for testing visual-navigation algorithms.

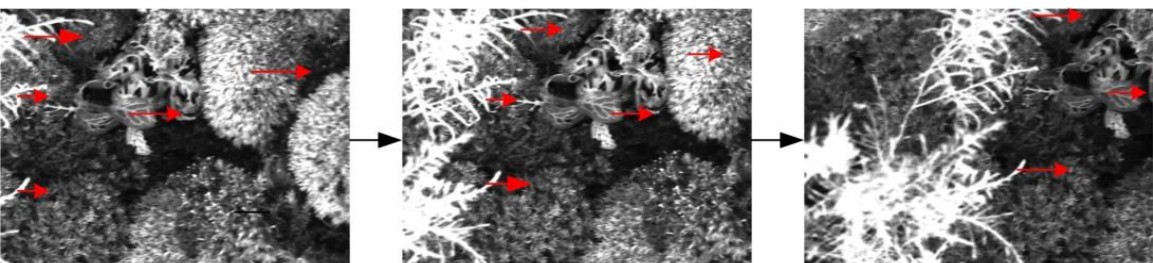

**Figure 20.** Optical-flow-direction diagram.

**6. Conclusions**

1. A semi-physical simulation platform for UAV visual navigation based on a real-time control system is proposed. The mechanical mechanism consists of a three-axis translation table and a three-axis rotary table, and the real-time control system is built based on the EtherCAT bus.
2. The repeatable positioning accuracy of the three-axis rotary table reached 0.006°, the repeatable positioning accuracy of the three-axis translation table reached 0.033 mm, and the dynamic-positioning accuracy reached 0.04° and 0.4 mm.
3. The six-axis position and angle information were converted into UAV flight animation using Qt and Open Inventor to achieve real-time trajectory tracking. The system can meet the simulation-testing requirements of the quadrotor UAV visual navigation, accelerate the optimization iterations of visual-navigation algorithms, and reduce the cost of outdoor-navigation-algorithm testing.

4.    There are many uncontrollable factors in UAV visual navigation. Therefore, it is necessary to conduct indoor semi-physical simulations before outdoor-navigation experiments. The platform can also support military-testing tasks, such as camouflage-target identification and dynamic target tracking.

5.    The limitation of this study is that due to the limitation of the mechanical structure, it is difficult to simulate a large turning angle of the UAV, and it is not possible to simulate a sudden sharp turn with high accuracy.

6.    For future research, there are two main aims. The first is to address the inability of existing mechanical structures to capture aerodynamic disturbances. This problem can be remedied by adding an aerodynamic-disturbances simulation model to the simulation computer. By combining real-time flight-attitude acquisition with mathematical models of aerodynamic disturbances, it is possible to achieve a more realistic simulation. The second is to apply the existing simulation platform to more scenarios, such as unmanned vehicles. Considering that the environment of the simulation platform is a miniature scenario, the model of unmanned vehicles under testing should also be scaled down. The mechanical structure at the end of the three-axis rotary table needs to be redesigned to adapt to the needs of the unmanned-vehicle ground simulation.

**Author Contributions:** Conceptualization, Z.L. and W.W.; methodology, Z.L. and W.W.; software, Z.L. and Y.L.; validation, X.Z.; formal analysis, X.Z.; investigation, H.W.; resources, Z.L.; data curation, X.Z.; writing—original draft preparation, Z.L.; writing—review and editing, Z.L., W.W. and Y.L.; visualization, Y.L.; supervision, T.Z. and X.W.; project administration, Z.L. and F.H.; funding acquisition, Z.L. and F.H. All authors have read and agreed to the published version of the manuscript.

**Funding:** This research was funded by Funding of the Natural Science Foundation of Fujian Province of China, grant number 2021J05113 and 2020J05101, Funding of the 2020 Fujian Province Young and Middle-aged Teacher Education Research Project (Technology), grant number JAT200030, Funding of the Fuzhou University Research Start-up Funding, grant number GXRC-20051, and Funding of the Crosswise Project of 'Research on the Air Pressure Simulation System Design of Aero-engine Surge', grant number 2021011902.

**Institutional Review Board Statement:** Not applicable.

**Informed Consent Statement:** Not applicable.

**Data Availability Statement:** The data presented in this study are available on request from the corresponding author. The data are not publicly available due to privacy.

**Acknowledgments:** The authors would like to thank Fuzhou University and Nanjing University of Aeronautics and Astronautics for their support. The authors are sincerely grateful to all the reviewers for their valuable comments.

**Conflicts of Interest:** The authors declare no conflict of interest.

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
