# Peer review of "Design and Experimental Study of a Novel Semi-Physical Unmanned-Aerial-Vehicle Simulation Platform for Optical-Flow-Based Navigation"

_aerospace, doi:10.3390/aerospace10020183_

Round 1

Reviewer 1 Report

Line 454-455, the cumulative positioning accuracy, is it really measured after 1000 meters (1 km), or should there be 1000 mm?

The same applies for table 4, page 20, where it is mentioned again.

Further comment from Reviewer.

The main part of the evaluation is covered in the completed evaluation form. However I can comment this: This manuscript is engineering / applied science type. It describes on step-by-step basis the design, construction and evaluation of a hardware platform with servo motors for indoor drone testing and development. There is not much novelty in theory (theoretical science) but rather in application of methods in engineering. The authors apply their theoretical and practical knowledge to the individual steps of the design and implementation of the simulation platform. Each step is described theoretically, its design is described and the results of its testing are presented. Finally, the evaluation of the practical tests is described and the achieved accuracy of the motion of the designed platform is evaluated. The proposed platform can be used for testing optical navigation systems of unmanned vehicles. The only limiting factor is its physical dimensions.

Reviewer 2 Report

The paper presents the development of a mechanical structure with 6 degrees of freedom that can accurately follow a prescribed trajectory.   The paper presents adequate experimental results supporting the high accuracy of the structure following prescribed trajectories.

The paper also discusses the use of the mechanical structure to simulate UAV vision-based navigation (lines 88-105).  It is my understanding that the mechanical structure is used to follow trajectories generated by UAVs.  I am wondering: What is the advantage of using the mechanical structure instead of using a real quadrotor whose position is measured accurately via a motion capture system?  Is it the case that state-of-the-art motion capture systems are less accurate in measuring the exact position of UAVs?  If yes, then by how much?

Also, is it of interest whether the mechanical structure captures aerodynamic disturbances?  I think currently such disturbances cannot be captured, and I am not sure if this compromises the realism of the achieved simulation.

All in all, in my opinion, I would elaborate on what is the practical advantage and purpose of the mechanical structure with respect to testing vision-based navigation algorithms.

Additional comments:

- I wasn't able to identify where Fig. 13 and Fig. 20 are discussed.  

- In the conclusions, it is stated "This study is a previous step in conducting the validation of the UAV outdoor visual 516 navigation algorithm".  I am not sure what the authors mean here by saying the "previous" step.

Round 2

Reviewer 2 Report

The authors addressed my comments.  Thank you